# Plasma Metabolomics Profile of “Insulin Sensitive” Male Hypogonadism after Testosterone Replacement Therapy

**DOI:** 10.3390/ijms23031916

**Published:** 2022-02-08

**Authors:** Lello Zolla, Marcello Ceci

**Affiliations:** Department of Ecological and Biological Sciences (DEB), University of Tuscia, 00110 Viterbo, Italy; m.ceci@unitus.it

**Keywords:** insulin sensitivity, hypogonadism, testosterone therapy, Cori cycle

## Abstract

Male hypogonadism is a disorder characterized by low levels of testosterone, but patients can either show normal insulin (insulin-sensitive (IS)) or over time they can become insulin-resistant (IR). Since the two groups showed different altered metabolisms, testosterone replacement therapy (TRT) could achieve different results. In this paper, we analyzed plasma from 20 IS patients with low testosterone (<8 nmol/L) and HOMAi < 2.5. The samples, pre- and post-treatment with testosterone for 60 days, were analyzed by UHPLC and mass spectrometry. Glycolysis was significantly upregulated, suggesting an improved glucose utilization. Conversely, the pentose phosphate pathway was reduced, while the Krebs cycle was not used. Branched amino acids and carnosine metabolism were positively influenced, while β-oxidation of fatty acids (FFA) was not activated. Cholesterol, HDL, and lipid metabolism did not show any improvements at 60 days but did so later in the experimental period. Finally, both malate and glycerol shuttle were reduced. As a result, both NADH and ATP were significantly lower. Interestingly, a significant production of lactate was observed, which induced the activation of the Cori cycle between the liver and muscles, which became the main source of energy for these patients without involving alanine. Thus, the treatment must be integrated with chemicals which are not restored in order to reactivate energy production.

## 1. Introduction

Male hypogonadism is a condition in which there is low testosterone production. Testosterone is a hormone that plays a key role in carbohydrate, fat, and protein metabolism [1]. The low levels of circulating plasma testosterone determine non-specific clinical signs and symptoms, such as dyslipidemia, elevated serum triglycerides, reduced high-density lipoprotein cholesterol (HDL), elevated fasting glucose levels, and high blood pressure [2]. These conditions predispose the body to developing other metabolic risk factors like metabolic syndrome and type 2 diabetes, which contribute to cardiovascular risk [3].

A hypogonadal state can start in normoinsulinaemic patients (IS patients), but over time, blood insulin concentration increases (IR patients), leading to further metabolic alterations and clinical complications, such as type 2 diabetes [4]. In our recent investigations, we distinguished normoinsulinaemic hypogonadic males and hyperinsulinaemic subjects through HOMAi, showing metabolomic differences between them [5,6].

In both patient groups, testosterone replacement therapy (TRT) is the most common treatment applied for improved symptoms of hypogonadism [7]. Treatment can be provided through injectable testosterone esters, transdermal testosterone (gels or patches), or oral testosterone in the form of testosterone undecanoate. All of these delivery modes are acceptable in appropriate doses and allow patients the benefit of having a variety of options to choose from [7]. TRT improves their sense of wellbeing, sexual function, mood, libido, bone density, muscle bulk, and muscle strength. It can also decrease visceral and peripheral body fat and reduce insulin resistance and blood sugar [8,9]. Large observational registry studies of long-term TRT in men with testosterone deficiency have reported consistent and sustained weight loss and decreased weight conference and BMI [2,10].

Although TRT is performed for long time periods, some clinics suggest that monitoring should occur every 3 to 6 months in the first year and at yearly intervals thereafter if the patient is stable [8]. Despite the improvement in the clinical conditions that testosterone therapy imports onto hypogonadal patients, which is visible only after a long time, little is known about the effect of testosterone on the metabolic and lipidomic pathways that could better help clinics to evaluate doses and time of treatment. In this regard, at the end of the treatment, beyond measuring the testosterone value and revealing clinical parameters, such as fatigue, libido, reduced energy, depressive symptoms, etc., an extensive analysis of a significant number of metabolites in the blood would confirm how tissues responded to testosterone recovery. In particular, it is essential to examine these hypogonadal patients with their insulin still active, in order to identify which metabolisms are reactivated before that insulin resistance occurs and, therefore, to understand how the synergy or antagonism between the two hormones coexists and how an endocrinologist should intervene. The “Homic” strategy, especially metabolomics, could help to explore the mechanisms underlying the metabolic alterations. Metabolomics, based on mass spectrometry, is the study of small low-molecular-weight metabolites in complex biological samples, aiming to characterize and quantify the small molecules in such samples [11]. Metabolomic analysis can detect altered metabolic responses given by TRT, improving drug treatment efficacy and safety monitoring [12]. Since it is our hypothesis that IS and IR patients respond differently to the same testosterone therapy, we performed plasma metabolomics analyses on both patients separately, and monitored which metabolisms were improved after TRT.

In this paper, we investigate plasma metabolites of hypogonadal men showing normal values of insulin (IS patients), testosterone concentration (lower < 8 nmol/L), HOMAi < 2.5, and BMI (25.44 ± 3.04) pre- and post-treatment with testosterone for 60 days. All patients showed testosterone levels similar to that of the controls after 40 days of treatment, but we decided to perform blood collection after 20 days in order to ensure a complete restoration and equilibration of all metabolic pathways. Analysis revealed that several pathways were positively influenced by TRT, such as glycolysis, but not the pentose phosphate pathway and the Krebs cycle. Branched amino acids and carnosine metabolism were positively activated, while glycerol and malate–aspartate shuttle reduced. As a result, low NADH and ATP values were recorded. Nevertheless, a significant production of lactate was observed, which activated the Cori cycle between liver and muscles without involving alanine, which decreased significantly. In the absence of the Krebs cycle and soft glycolysis, this seems to be the main source of energy for these patients, after TRT. Thus, the restoration of testosterone values does not seem to reactivate all metabolisms, but something must be restored and better investigated. Muscle and adipose tissue are benefited, as well as collagen synthesis. Fortunately, the inhibition of glutaminolysis was observed. For other metabolisms, chemical supplements must be added to the therapy.

## 2. Results

In this study, we analyzed 20 subjects in the control group and 20 normo-insulinemic hypogonadal males for 60 days, the latter having received testosterone before and after the study. Table 1 provides the demographic and clinical chemistry data for the subjects in the study. It is of note that these IS hypogonadal patients showed a restored testosterone in the medium-normal range after just 40 days of treatment (data not shown), but blood collection was performed after 60 days when the level of testosterone was about 19.20 ± 9.10 nmol/L in order to ensure complete metabolic pathway restoration and equilibration. Using plasma from these patients, we conducted an exploratory metabolomics analysis using non-targeted, high-resolution mass spectrometry. Comparison of metabolomics before and after testosterone therapy allowed us to detect which metabolic pathways were restored upon testosterone addition. To better identify which metabolic pathways were most affected in hypogonadal patients after TRT, we performed a functional enrichment analysis of the experimental data with MetaboAnalyst 3.0.

Plasma metabolites together with their relative concentrations were analyzed using an MSEA (Figure 1A). This approach identifies biologically significant models that are significantly enriched in quantitative metabolomic data. In MSEA, the *p*-value indicates the strength of the association between the profiled metabolite and the class label. Figure 1A shows in red the most affected metabolic pathways related to post-hypogonadism treatment, while those altered to a lesser extent are in orange. It might be observed that the most significant metabolites identified were β-oxidation of very long fatty acids, oxidation of branched chain fatty acids, pentose phosphate pathway, and gluconeogenesis. In parallel, we utilized the Pathway Analysis module of MetaboAnalyst, which combines results from the pathway enrichment analysis with the pathway topology analysis to identify the most relevant pathways altered (Figure 1B). Significantly altered pathways (*p* < 0.05) that also had high impact values included valine isoleucine leucine biosynthesis, pentose phosphate pathway, citrate cycle, alanine aspartate metabolism, and glycolysis. 

Table 2 and Appendix A, placed in the supplementary data, lists the metabolites that were selected for later discussion, showing their percentage increase or decrease before and after treatment of the hypogonadal patient with testosterone.

Glycolysis was significantly upregulated (Figure 2A), suggesting that testosterone improved glucose utilization. Interestingly, after TRT, there was a significant lactate production of 95% (Figure 2A). Conversely, the pentose phosphate pathway (PPP), which was upregulated in hypogonadism IS, was reduced after treatment (Figure 2B). Interestingly, glutathione disulphide (GS-SG) decreased to 96% (Figure 2C), indicating a lower oxidative stress. However, a smaller amount of total NADH was produced (Figure 2C). Moreover, as shown in Figure 2D, glyceraldehyde 3-phosphate produced more glycerol 3-phosphate (25%), and consequently the glycerol shuttle was downregulated, as also supported by the low levels of dihydroxyacetone (50%) and NAD (40%) (Figure 2C). Thus, glycerol 3-phosphate was consumed preferentially to lipid synthesis, reacting with fatty acids to produce more glycerophospholipids and phosphatidylcholine (PC) (Figure 2C, right side), as better documented in our recent manuscript [13].

As a result of increased glycolysis, the level of acetyl-CoA was up-produced to 90% (Figure 3). A small portion was converted into mevalonic acid (15%) (Figure 3A) and then into cholesterol, which did not increase after a short time, after 60 days of treatment (Table 1), but at a longer time (data not shown). Moreover, since the entrance of acetyl-CoA and fatty acids into the mitochondria is mediated by acetylcarnitine, the lower concentration of the latter (Figure 3B), inhibited both transportations and their utilization by β-oxidation. Krebs cycle (TCA) remained lower (Figure 3), and consequently ATP levels were low (Figure 3C), as well as NADH production, indicating that testosterone therapy did not resolve the energy supply through canonical pathways. 

Furthermore, the downregulation of glutaminolysis produced glutamate accumulation, which switched off the malate–aspartate shuttle (Figure 4A), which is the main energy support recorded in IS hypogonadism before TRT [5]. Regarding amino acid metabolism, aside from the accumulation of glutamate (95%), a decreased branched-chain amino acids utilization was recorded, such as valine (30%), leucine, and isoleucine (40%) (Figure 4B). However, most amino acids decreased significantly (Figure 4C) including alanine (90%).

Finally, Figure 5 shows that carnosine (30%) as well as its precursors histidine (15%), β-alanine, and uracil (10%) were restored after TRT, becoming similar to the levels of the controls. 

## 3. Discussion

Recent clinical practice guidelines recommend testosterone replacement therapy (TRT) for adult men with low testosterone levels (hypogonadal men) to improve symptoms and increase testosterone levels in the medium–normal range [14]. However, the efficiency of androgen replacement therapy is controversial, as it was observed in some hypogonadism patients but not in all [15], reporting that after the restoration of testosterone levels, sexual desire, activity, and frequency of erections increased [16] and anthropometric parameters improved [2]. Thus, a better knowledge of which metabolic pathways were restored with only testosterone therapy can help to improve knowledge of how to reach a complete metabolomic restoration in all patients.

Metabolomic analysis performed here on IS hypogonadism patients revealed that after 60 days of TRT, not all metabolic pathways were restored (Figure 1). Moreover, most energy came from a reactivated glycolysis or probably from another source, because the Krebs cycle was not improved and the pentose phosphate pathway decreased, in agreement with what was observed in testicular feminized mice (Tfm) [1]. In fact, hepatic glucose-6-phosphatedehydrogenase (G6PD), the key enzyme of PPP, was elevated compared to XY mice (*p* < 0.001), but testosterone treatment reduced its expression. On the contrary, GSSG decrease significantly, indicating that TRT reduces oxidative stress, in agreement with Mancini and colleagues, who have shown that hypogonadism could represent a condition of oxidative stress and that TRT reported seminal total antioxidant capacity (TAC) values at the same levels observed in normogonadic patients [17]. Indeed, testosterone has been shown to increase the expression of GLUT4 in cultured skeletal muscle cells, hepatocytes, and adipocytes [18,19] as well as augmenting membrane translocation and promoting glucose uptake in adipose and skeletal muscle tissue [20]. Thus, both muscles and adipose tissues should benefit from testosterone restoration. Interestingly, lactate increased significantly after TRT, in agreement with Burns [21]. In fact, there is a correlation between lactate and testosterone production in rat Leydig cells, [22] and in these patients the testosterone production was stimulated by lactate and vice versa. Thus, lactate accumulation might not imply a state of anaerobic glycolysis but simply a state of accelerated glycolysis (glycolytic flux is higher than the tricarboxylic acid cycle flux) [23], and/or higher lactate production. In this regard, Enoki and colleagues underline the correlation between testosterone and lactate, showing that testosterone induces an increase in monocarboxylate transporters of lactate in rat skeletal muscle [24].

After TRT, the level of acetyl-CoA increased significantly (Figure 3). This was also recorded for TRT of IR hypogonadism (Zolla et al.; Plasma metabolomics profile of testosterone replacement therapy in insulin-resistance male hypogonadism IJMS, 2021, submitted), which clearly indicates that testosterone administration induced metabolic shifts, although different, to achieve an increase of acetyl-CoA. Part of acetyl-CoA was converted to mevalonic acid and then into cholesterol synthesis, but cholesterol level in all patients did not increased significantly after 60 days (Table 1) but after longer time, supporting the fact that short periods of treatment were insufficient for restoring all lipid metabolisms [25]. In agreement, our recent papers on lipidomic profiles [13] showed that after 60 days of testosterone treatment, the altered sphingomyelins (SM), phosphatidilcoline (PC), and phospholipase C (LPC) were completely restored to control levels, while other lipid derivates such as LDL, triglycerides, cholesterol, and HDL did not change significantly. It is of note that in the latter paper, all lipids were measured by mass spectrometry and not by separate biochemical assays, allowing us to have a simultaneous overview of all classes of lipids. More fatty acids (FFA) were produced from an increased concentration of acetyl-CoA, which reacting with 3-glycerol, produced more phosphoglycerides and phophatidylcholine (Figure 2D), but switched off the glycerol shuttle with a decrease of NADH production (Figure 2D). This is in agreement with Zaima et al. [26], who found that testosterone levels were upregulated in mice that received fish oil with high eicosapentaenoic acid content. In the testicular interstitial, eicosapentaenoic acid containing phosphatidylcholine was distributed, characteristically demonstrating the involvement of eicosapentaenoic acid in testosterone metabolism.

After TRT, lower levels of acetyl-carnitine were recorded (Figure 3B), which is fundamental in transporting fatty acids from the cytoplasm to mitochondria for β-oxidation. Consequently, more triglycerides were produced. As a result, a lipid accumulation might occur with a consequent decline in the availability of energy in the heart, skeletal muscles, and kidneys [20]. Decreased free carnitine levels were also recorded in late-onset hypogonadism (LOH) patients [27]. Interestingly, several studies emphasize the effect of carnitine as a replacement therapy for the treatment of hypogonadism to improve male reproductive function. These results showed the positive role of carnitine on sex hormones and the reproductive system, making carnitine an appropriate candidate for the therapy of symptoms associated with aging [28,29].

Finally, although more acetyl-CoA was available in IS hypogonadism after TRT, the Krebs cycle was not upregulated (Figure 3). A downregulation of the TCA cycle after TRT was also reported by Petersson et al. [30], who found that transdermal testosterone therapy for 6 months did not change the expression of PGC1α or genes involved in oxidative phosphorylation (OxPhos), TCA cycle, or lipid metabolism in the skeletal muscle of elderly men with subnormal bioavailable testosterone levels. They have shown that testosterone therapy has no effect on the protein levels of representative OxPhos subunits or on the protein abundance and phosphorylation of two enzymes, AMPK and p38 MAPK, known to regulate mitochondrial biogenesis through PGC1α [30]. Moreover, our data show that the Krebs cycle was not restored by glutaminolysis (Figure 3C), as previously observed in IS hypogonadism [5], before TRT. Thus, glutaminolysis was stopped, and more glutamate was available, blocking the malate–aspartate shuttle and the production of alanine and α-chetoglutarate, which decreased. Glutaminolysis is a metabolic reprogram in this disease to produce energy. In fact, glutamine can be metabolized in the TCA cycle, producing carbon and nitrogen, to generate energy and produce intermediates for the synthesis of the macromolecule. Unfortunately, this extreme adaptation was observed in tumour cells to receive energy for growth and uncontrolled cell proliferation [31,32]. Li et al. proposed the inhibition of glutaminolysis to stop the proliferation of colon cancer cells [33]. Furthermore, glutaminolysis in β-pancreatic cells is regulated by glucose. Glucose upregulation determines the inhibition of glutaminolysis [34], as observed here. Therefore, although testosterone does not directly improve TCA, it has a beneficial outcome because it upregulates glycolysis and consequently blocks glutaminolysis. As a consequence of the deactivation of glutaminolysis, a strong reduction of ATP production (Figure 2C) was observed, in agreement with Ponce, who showed a statistically non-significant improvement in energy [35].

However, if TRT was efficacious in restoring normal testosterone and some metabolisms, its benefit is not clear regarding its energy source, since both ATP and NADH levels were still lower. On the other hand, after TRT there was a reactivation of GLUT4 in muscles that ensured a restart of glycolysis, then the increased lactate may have induced the activation of the Cori cycle (Figure 6), where alanine was excluded from the cycle. Thus, lactate become the main energy source in highly oxidative cells (e.g., heart, brain, and lung) or in turn to be converted into glucose in the liver and kidney and used by muscles. Participation of glucose was possible, because in these patients insulin is active and glucose can be used. 

Regarding muscles, skeletal protein catabolism was reduced, and less branched amino acids such as leucine, isoleucine, and valine were released into the blood. It is of note that branched amino acids account for nearly 35% of the essential amino acids in muscle proteins. Their presence in the blood is indicative of a higher skeletal-muscle protein catabolism to produce energy in hypogonadism [6], causing a decrease in muscle mass. This agreed with D’Antona [36], who demonstrated the anti-aging role of the BCAAs leucine/isoleucine and valine in mitochondrial biogenesis in mammals. These results confirmed that the testosterone does not improve the function of mitochondria and the mitochondrial biogenesis characteristics of aging [37]. In this regard, carnosine and its intermediates increased, supporting the role of testosterone in the control of muscle protein synthesis [38,39]. Higher concentrations of proline and lysine were also recorded, suggesting a positive influence of testosterone in the synthesis of collagen fibres (Figure 4E), reducing bone loss and excretion of bone-degradation parameters, such as hydroxyproline, as reported by Tenover [40].

## 4. Materials and Methods 

### 4.1. Patients Samples: Study Design and Participants

Human blood plasma samples were collected in accordance with ethical guidelines and approved standard clinical protocol after overnight fasting. EDTA-plasma was prepared by 10 min centrifugation at 4 °C and 3000 g. We evaluated 20 hypogonadal male patients and 20 age- and BMI-matched controls (Table 1). All subjects enrolled were informed about the study protocol and gave their written consent. The diagnosis of hypogonadism was based on the presence of clinical symptoms related to this condition (e.g., delayed sexual development, reduced libido, or erectile dysfunction) and on the results of standard hormonal exams (total testosterone < 8 nmol/L). The patients affected by hypogonadism were included only if they had HOMAi < 2.5. The participants of the control group were healthy males who were referred to the Outpatient Clinic of Endocrinology and Metabolism for check-up. As shown in Table 1, no differences were found in the baseline characteristics between groups.

### 4.2. Study Treatment

The study was carried out at Fondazione Policlinico Gemelli IRCCS, Rome, Italy. Twenty healthy men and twenty men with hypogonadism, diagnosed using both clinical symptoms of hypogonadism, including erectile dysfunction, decreased libido, and/or decreased energy, as well as evidence of low serum T (≤8 nmol/L), were enrolled in our study. The hypogonadic patients were treated with testosterone preparation gel 2%. The gel was formulated to have a similar application and appearance. Serum testosterone concentration was measured at 60 days. All patients gave their informed consent before participating in the study.

Metabolites were extracted by adding 200 µL of each plasma sample to 600 µL of cold (−20 °C) chloroform:methanol:water (1:3:1 ratio). Samples were vortexed for 1 min and left on ice for 2 h for complete protein precipitation. The solutions were then centrifuged for 15 min at 15,000× *g*.

### 4.3. UHPLC-HRMS

Twenty microliters of extracted plasma were injected into an Ultra High Performance Liquid Chromatography (UHPLC) system (Ultimate 3000, Thermo, Waltham, MA, USA) and performed in positive mode: Samples were loaded onto a Reprosil C18 column (2.0 mm × 150 mm, 2.5 μm—Dr Maisch, Munich, Germany) for metabolite separation. Chromatographic separations were achieved at a column temperature of 30 °C and a flow rate of 0.2 ml/min. For positive-ion mode (+) MS analyses, a 0–100% linear gradient of solvent A (ddH2O, 0.1% formic acid) to B (acetonitrile, 0.1% formic acid) was employed over 20 min, returning to 100% A for 2 min and a 6-min post-time solvent A hold. Acetonitrile, formic acid, and HPLC-grade water and standards (≥98% chemical purity) were purchased from Sigma Aldrich. The UHPLC system was coupled online with a mass spectrometer Q Exactive (Thermo) scanning in full MS mode (2 μscans) at 70,000 resolution in the 67 to 1000 m/z range, with a target of 1106 ions, maximum ion injection time (IT) of 35 ms, 3.8 kV spray voltage, 40 sheath gas, and 25 auxiliary gas, operated in positive-ion mode. Source ionization parameters were spray voltage, 3.8 kV; capillary temperature, 300 °C; and S-Lens level, 45. Calibration was performed before each analysis against positive- or negative-ion-mode calibration mixes (Piercen et, Thermo Fisher, Rockford, IL, USA) to ensure sub-ppm error of the intact mass. Metabolite assignments were performed using computer software (Maven, 18, Princeton, NJ, USA), upon conversion of raw files into mzXML format through MassMatrix (Cleveland, OH, USA).

### 4.4. Metabolomic Data Processing and Statistical Analysis

Raw files of replicates were exported, converted into mzXML format through MassMatrix (Cleveland, OH, USA), and then processed by MAVEN software (http://maven.princeton.edu/, accessed on 7 January 2022). Mass spectrometry chromatograms were elaborated for peak alignment, matching and comparison of parent and fragment ions, and tentative metabolite identification (within a 2-ppm mass-deviation range between observed and expected results against the imported KEGG database). To further explore the metabolic differences between the two groups of subjects, multivariate statistical analyses were employed on an MS data set consisting of 15 hypogonadal men pre and post treatment. Multivariate statistical analyses were performed on the entire metabolomics data set using the MetaboAnalyst 3.0 software, San Diego, CA, USA, which also served to overview the data variance structure in an unsupervised manner. Before the analysis, raw data were normalized by median and auto-scaling to increase the importance of low-abundance ions without significant amplification of noise. The web-based tools MSEA (Metabolite Set Enrichment Analysis) and MetPA (Metabolomic Pathway Analysis), which are incorporated into the MetaboAnalyst platform, were used to perform metabolite enrichment and pathway analyses, respectively. The data for the metabolites detected in all samples were submitted into MSEA and MetPA with annotations based on common chemical names. Accepted metabolites were verified manually using HMDB, KEGG, and PubChem DBs. A Homo sapiens-pathway library was used for pathway analysis. Global test was the selected pathway enrichment analysis method, whereas the node importance measure for topological analysis was the relative betweenness centrality. For MSEA metabolites, data were mapped according to HMDB, and the “metabolite pathway associated metabolites set” library (currently 88 entries) was chosen for the enrichment analysis, which was performed using the package global test. Results were graphed with Graphpad Prism 5.01 (Graphpad Software Inc., Edmonton, AB, Canada). Statistical analyses were performed with the same software. Data are presented as mean ± SEM of fold-change relative to the metabolite levels in controls. Differences were considered statistically significant at * *p* < 0.05 and further stratified to ** *p* < 0.01 and *** *p* < 0.001, respectively.

## 5. Conclusions

The overall metabolic profile of blood in IS hypogonadism after 60 days of TRT revealed that testosterone alone is not sufficient to re-establish all metabolisms, apart from muscles and adipose tissue that showed benefits. Regarding energy, glucose upregulation determines the inhibition of glutaminolysis [34] which in some cases is a promoter of cancer development [31,33]

However, in these patients the presence of still active insulin allows the activation of the Cori cycle to supply energy and avoids the production and accumulation of ketone bodies and their negative effects, as revealed in insulin-resistant patients (Zolla et al.; Plasma metabolomics profile of testosterone replacement therapy in insulin-resistance male hypogonadism IJMS, 2021, submitted). However, the addition to TRT of other chemicals could better restore both glucose and lipid metabolisms, such as carnitine. Its supplementation has been shown to ameliorate aging-related sexual dysfunction, inhibit the development of cardiovascular disease, and reduce levels of free fatty acids [27]. Carnitine or acetyl-carnitine could act together with testosterone to prevent disease progression and improve the condition of hypogonadism patients. The addition of citrate and amino acids could help.

## 6. Patents

This study was approved by the local ethics committee (Università Cattolica del Sacro Cuore, Rome), protocol P/ 740/CE/2012. A written informed consent was provided by all the subjects.

## Figures and Tables

**Figure 1 ijms-23-01916-f001:**
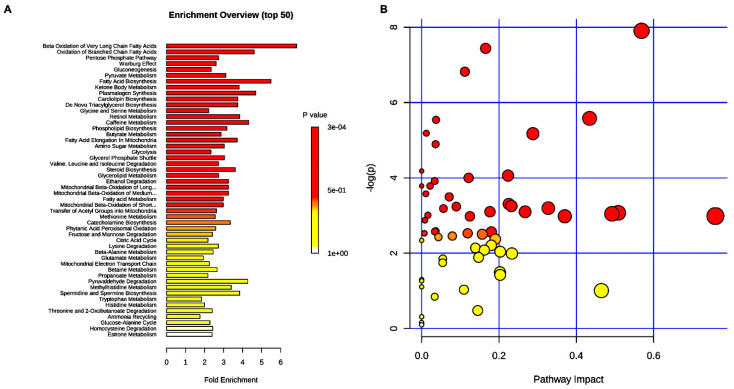
(**A**) Metabolic Set Enrichment Analysis (MSEA) showing the most altered metabolites revealed in the plasma of hypogonadal men before and after testosterone replacement treatment (TRT). Color intensity (white-to-red) reflects increasing statistical significance, while the circle diameter covaries with pathway impact. The graph was obtained by plotting on the *y*-axis the –log of *p*-values from pathway enrichment analysis and on the x-axis the pathway impact values derived from pathway topology analysis. (**B**) Metabolic Pathway Analysis (MetPA). All the matched pathways are displayed as circles. The color and size of each circle are based on the *p*-value and pathway impact value, respectively. The graph was obtained by plotting on the *y*-axis the −log of *p*-values from the pathway enrichment analysis and on the *x*-axis the pathway impact values derived from the pathway topology analysis.

**Figure 2 ijms-23-01916-f002:**
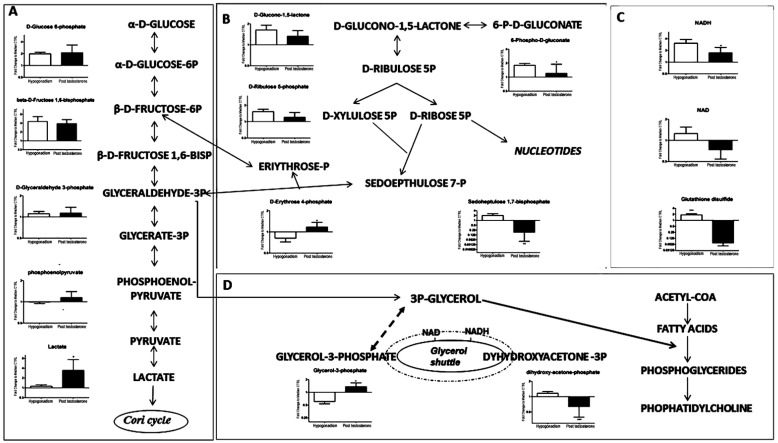
Metabolomic profile of glucose metabolism. (**A**) All glycolytic intermediates were upregulated after TRT. (**B**) Intermediates of pentose phosphate pathway were restored to that similar to control. (**C**) Decrease of glutathione disulphide as a marker of the improvement of oxidative stress. Deregulation of level of NAD and NADH. (**D**) The glycerol shuttle was not active to contribute to the oxidative phosphorylation pathway in the mitochondria. It was used for phospholipid synthesis. All data are shown as mean ± SEM of fold-change relative to the metabolite levels in controls. * *p* < 0.05.

**Figure 3 ijms-23-01916-f003:**
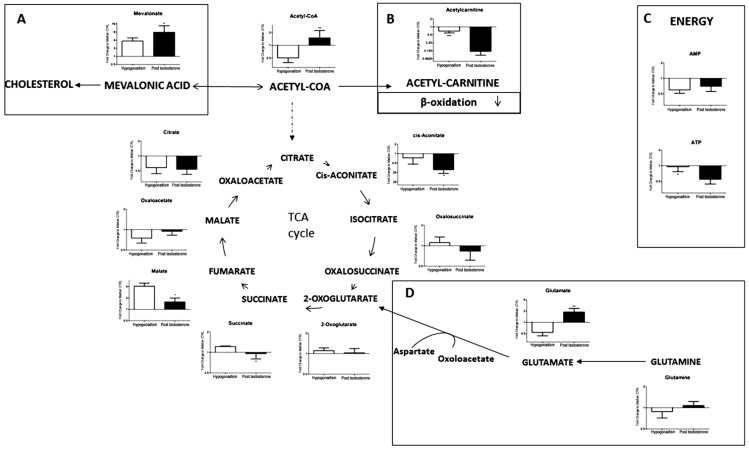
Metabolism involved in acetyl-CoA catabolism. Intermediates of Tricarboxylic acid (TCA) cycle measured in the plasma of hypogonadal patients. We found an overall decreased level of TCA-cycle metabolites. (**A**) Level of mevalonic acid increased after therapy, yet the concentration of cholesterol did not improve. (**B**) Acetyl-carnitine and fatty acid oxidation are significantly reduced after TRT. (**C**) Levels of energy metabolites production AMP and ATP. (**D**) Intermediates of the glutaminolysis pathway. This stepwise became downregulated after TRT. All data are shown as mean ± SEM of fold-change relative to the metabolite levels in controls. * *p* < 0.05, ** *p* < 0.01, *** *p* < 0.001.

**Figure 4 ijms-23-01916-f004:**
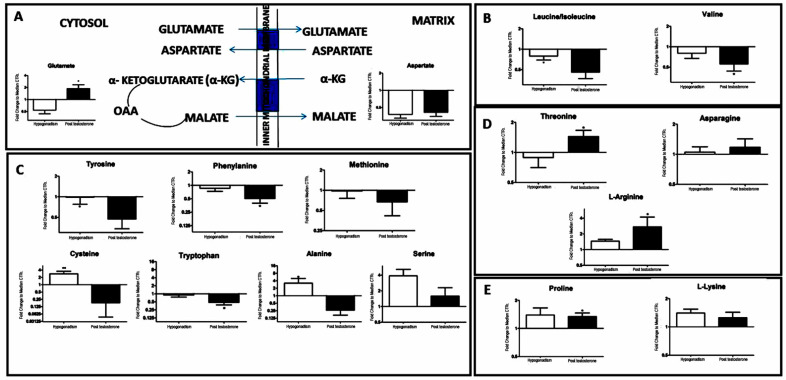
(**A**) The downregulation of glutaminolysis led to a stop of the activation of the malate aspartate cycle, as shown by the levels of glutamate and aspartate. (**B**) Decreased levels of branched-significantly decreased. (**C**) Plasma amino acids that were significantly decreased. (**D**) Plasma amino acids that were significantly increased. (**E**) Restoration of proline and lysine involved in collagen fibers formation. All data are shown as mean ± SEM of fold-change relative to the metabolite levels in controls. * *p* < 0.05.

**Figure 5 ijms-23-01916-f005:**
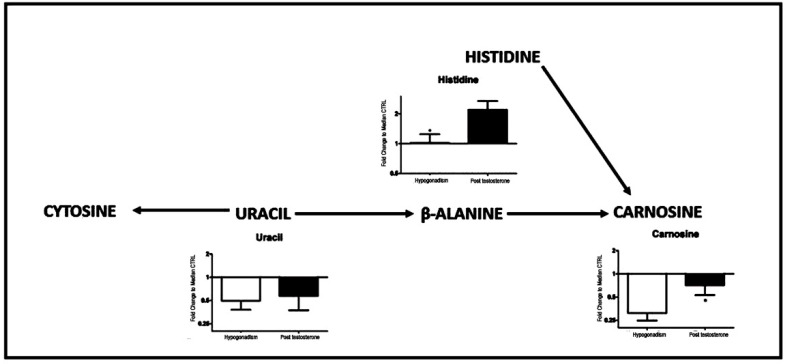
Schematic model summarizing change in carnosine metabolism. The carnosine production from β-alanine increased in response to testosterone therapy. All data are shown as mean ± SEM of fold-change relative to the metabolite levels in controls. * *p* < 0.05.

**Figure 6 ijms-23-01916-f006:**
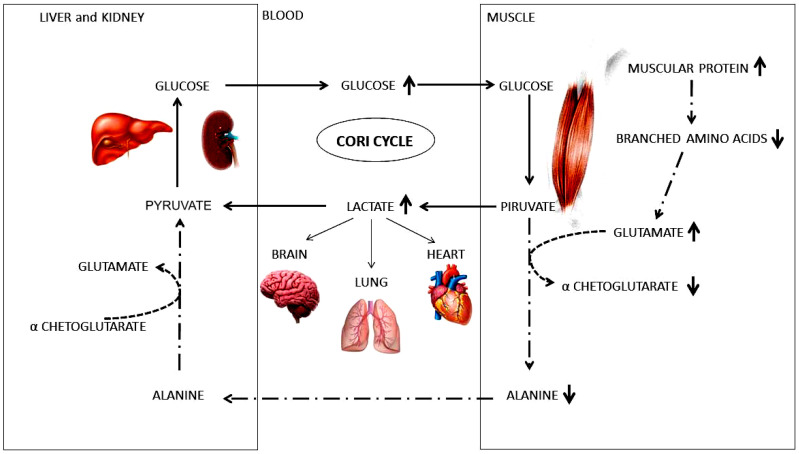
Overview of the energy production in hypogonadic insulin-sensivity male after TRT. Pyruvate produced by glycolysis is converted into lactate rather than alanine. Lactate can both be directly used as main energy source in highly oxidative cells such as brain, lung, heart or enter in liver or kidney and converted in glucose, that is used to produce energy (Cori Cycle).

**Table 1 ijms-23-01916-t001:** Characteristics of Study Participants. BMI: body mass index; TG: triglycerides; LDL: low- density lipoproteins; HDL: high-density lipoproteins. Data are presented as the mean ± SD. Statistical Post-Hoc Analysis was performed with Tukey’s Test ** *p* < 0.01, *** *p* < 0.001.

	Control (A)	Hypogonadic IS (B)	IS after TRT (C)	*p*-Value	Tukey HSD *p*-Value
**Subjects**	n-20	n-20	n-20	-	-
**Age**	42.54 ± 13.67	42.18 ± 16.01	42.18 ± 16.01	-	-
**BMI (Kg/m^2^)**	24.91 ± 4.01	25.44 ± 3.04	25.24 ± 3.09	0.93	-
**Testosterone (nmol/L)**	20.02 ± 7.47	6.35 ± 4.35	19.20 ± 9.10	0.0001 *** *p*	(A vs B) 0.001 ** *p* (B vs C) 0.001 ** *p*
**Glucose (mg/100 mL)**	84.72 ± 4.38	81.45 ± 12.51	86.90 ± 6.77	0.3	-
**Insuline (mUI/L)**	7.77 ± 3.13	6.72 ± 2.88	6.99 ± 2.98	0.69	-
**HOMAi**	1.79 ± 0.86	1.97 ± 0.67	1.47 ± 0.70	0.92	-
**Tg (mmol/L)**	96.36 ± 51.39	117.90 ± 62.73	125.54 ± 63.18	0.5	-
**Cholesterol (mmol/L)**	196.72 ± 29.18	212.81 ± 42.6	210.18 ± 53.18	0.6	-
**HDL Cholesterol (mmol/L)**	53.90 ± 11.98	52.63 ± 15.01	47.45 ± 15.47	0.54	-
**LDL Cholesterol (mmol/L)**	133.45 ± 33.07	136.36 ± 38.74	127.36 ± 44.31	0.85	-

**Table 2 ijms-23-01916-t002:** List of metabolites selected to evaluate the effect of testosterone therapy and their percentage increase or decrease before and after treatment. The values are reported as percentual difference compared to control (not included in table). *Green*, increased molecules; *red* decreased molecules.

Molecule	M.W.	Hypog. (%)	Post Testost. (%)
D-Glucose 6-Phosphate (260.14)	260.14	100	105
beta-D-Fructose 1,6-bisphosphate (340.114)	340.114	350	300
D-Glyceraldehyde 3-phosphate (170.06)	170.06	5	10
Phosphoenolpyruvate (168.04)	168.04	−1	20
Lactate (90.08)	90.08	10	280
D-Glucono-1,5-lactone (178.14)	178.14	70	30
D-Ribulose 5-phosphate (230.11)	230.11	60	15
D-Erythrose 4-phosphate (200.084)	200.084	−20	−98.9
6-P-D-Gluconate (276.135)	276.135	90	10
Sedoheptulose 1,7-bisphosphate (370.14)	370.14	100	−75
NADH (663.43)	663.43	280	80
NAD (663.43)	663.43	10	−20
Glutathione disulfide (610.6)	610.6	98	−90
Glycerol-3-Phosphate (172.074)	172.074	−40	20
Dyhydroxyacetone-3P (170.06)	170.06	5	−30
Mevalonate (148.16)	148.16	290	700
Acetyl-CoA (809.57)	809.57	−48	20
Acetyl-carnitine (203.236)	203.236	−20	−90
Citrate (192.124)	192.124	−99.6	−99.55
Oxaloacetate (132.07)	132.07	−18	−1
Malate (134.0874)	134.0874	300	25
Succinate (118.09)	118.09	10	−5
2-oxoglutarato (146.11)	146.11	4	1
Oxalosuccinate (190.11)	190.11	5	−15
cis-Aconitate (174.108)	174.108	−20	−100
Glutamate (147.13)	147.13	−48	90
Glutamine (146.14)	146.14	−5	2
AMP (347.2212)	347.2212	−40	−20
ATP (507.18)	507.18	−1	−60
Aspartate (133.11)	133.11	-60	−55
Leucine/isoleucine (131.17)	131.17	−15	−47
Valine (117.15)	117.15	−15	−40
Tyrosine (181.19)	181.19	−1	−48
Phenylanine (165.19)	165.19	−5	−48
Cysteine (121.16)	121.16	200	−85
Tryptophan (204.23)	204.23	−2	−50
Methionine (149.21)	149.21	−1	−25
Alanine (89.09)	89.09	280	−55
Serine (105.09)	105.09	299	60
Threonine (119.1192)	119.1192	−5	50
Asparagine (132.12)	132.12	−97	−90
L-Arginine (174.2)	174.2	60	200
Proline (115.13)	115.13	50	47
L-Lysine (146.19)	146.19	50	25
Histidine (155.1546)	155.1546	−99	110
Uracil (112.09)	112.09	−48	−40
Carnosine (226.3)	226.3	−55	−18

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
