# Peer review of "Plasma Metabolomics Profile of “Insulin Sensitive” Male Hypogonadism after Testosterone Replacement Therapy"

_ijms, 2022, doi:10.3390/ijms23031916_

Round 1

Reviewer 1 Report

  1. What are the conclusions in Abstract. The authors only summarized the results without making conclusive descriptions in the final.
  2. The image resolution of figures especially Figure 2 to 5 should be improved.
  3. The marks for statistical significance should be consistently fixed at the same group.
  4. In Figure 6, wrong spelling, PYRUVATE NOT PIRUVATE.

Author Response

Referee 1
Thanks for the constructive suggestions.
Unfortunately, the abstract can contain 200 words and it is not easy write all. 
However, I added a concluding sentence at the end which is then taken up again in the conclusions.
Fig. 2 and Fig. 5b have been improved.

Reviewer 2 Report

Herein, the authors analysed plasma from 20 IS patients with low testos- 13 terone (< 8 nmol/L) and HOMAi < 2.5. The samples, pre- and post-treatment with testosterone for 14 60 days, were analysed by UHPLC and mass spectrometry

Major issues

  1. I don’t found the results of UPLC-HRMS, a table summarizing all the components found their molecular weight and percentages in each collected samples should be added

Minor issues

  1. The introduction lacks a paragraph highlighting the novelty of the study should be addressed in the introduction section
  2. The major structures should be drawn using chemdraw or any other software
  3. The manuscript contains some spelling, grammatical and formatting mistakes that should be revised carefully
  4. The references should be carefully checked to be all in the same style.

Author Response

Referee 2
Thanks for the constructive suggestions.
UNTARGETED analyses carried out in mass spectrometry reveal thousands and thousands of chemical 
components, identified and valued in quantity. Subsequently with software programs, in our case Pathway 
Analysis module of MetaboAnalyst, we identify those increased or decreased, with respect to the control, 
which belong to the same biological cycle, eg. glycolysis, cyclo Kebs, etc, to show which metabolomic 
cycles are decreased or increased. This is shown in Fig. 1, which reveals metabolisms have been negatively 
or positively influenced by testosterone treatment, as in our case.
The results are then expressed in fold change, i.e. the ratio between the quantity of each component with 
respect to the control. Therefore, in the figures the horizontal line at 1 value represents the control value and 
the columns the increase or decrease of the component, before and after treatment with testosterone.
However, we added, as suggested by the referee, a Table 2 listing only the components shown in the work, in 
order to better clarify the variations of the components analysed.
- In the introduction three paragraphs have been added to highlighting the novelty of the study.
- In the text, we have written only the chemical components and not chemical structures.
Spelling, grammatical and formatting mistakes have been revised as well as references

Reviewer 3 Report

Comment on Manuscript “Plasma metabolomics profile of “insulin sensitive” male hy-2 pogonadism after testosterone replacement therapy” submitted to International Journal of Molecular Sciences

Comments to the Authors

This is a well-written and interesting manuscript. The authors report a plasma metabolomics analysis to study the effect of testosterone therapy on the metabolic and lipidomic pathways in IS and IR patients.

Although this is an enjoyable and pretty read, I would like to share my opinions and suggestions about the paper below, mainly for readers benefit:

  • The figures are too small and almost unreadable, please consider adjusting them
  • Overall, in the manuscript there is an issue with some words that are truncated with a hyphen (e.g. page 1- line 28: testos-terone, page 1- line 28: circu-lating, and many many more…)
  • In Table 1, the comma is used a decimal separator, while in the manuscript the dot is used. Please, use a consistent notation.
  • In Table 1, the authors report mean +- SD, while in Figure 2(and so on) they considered SEM, namely the standard error of the mean. Please could you explain the different choice of the reported variability?
  • I have wondered the data cleaning processes of authors. They should write their data cleaning processes, if any, in detail.

Author Response

Thanks for the constructive suggestions.
UNTARGETED analyses carried out in mass spectrometry reveal thousands and thousands of chemical 
components, identified and valued in quantity. Subsequently with software programs, in our case Pathway 
Analysis module of MetaboAnalyst, we identify those increased or decreased, with respect to the control, 
which belong to the same biological cycle, eg. glycolysis, cyclo Kebs, etc, to show which metabolomic 
cycles are decreased or increased. This is shown in Fig. 1, which reveals metabolisms have been negatively 
or positively influenced by testosterone treatment, as in our case.
The results are then expressed in fold change, i.e. the ratio between the quantity of each component with 
respect to the control. Therefore, in the figures the horizontal line at 1 value represents the control value and 
the columns the increase or decrease of the component, before and after treatment with testosterone.
However, we added, as suggested by the referee, a Table 2 listing only the components shown in the work, in 
order to better clarify the variations of the components analysed.
- In the introduction three paragraphs have been added to highlighting the novelty of the study.
- In the text, we have written only the chemical components and not chemical structures.
Spelling, grammatical and formatting mistakes have been revised as well as references

Round 2

Reviewer 2 Report

No additional comments

Reviewer 3 Report

The paper was improved and is now acceptable in its present form.